# A Pore Forming Toxin-like Protein Derived from Chinese Red Belly Toad *Bombina maxima* Triggers the Pyroptosis of Hippomal Neural Cells and Impairs the Cognitive Ability of Mice

**DOI:** 10.3390/toxins15030191

**Published:** 2023-03-03

**Authors:** Qingqing Ye, Qiquan Wang, Wenhui Lee, Yang Xiang, Jixue Yuan, Yun Zhang, Xiaolong Guo

**Affiliations:** 1School of Physical Education, Yunnan Normal University, Kunming 650500, China; 2Human Aging Research Institute, School of Life Science, Nanchang University, Jiangxi Key Laboratory of Human Aging, Nanchang 330031, China; 3Key Laboratory of Animal Models and Human Disease Mechanisms of the Chinese Academy of Sciences, Key Laboratory of Bioactive Peptides of Yunnan Province, Kunming Institute of Zoology, Chinese Academy of Sciences, Kunming 650201, China; 4Center for Excellence in Animal Evolution and Genetics, Chinese Academy of Sciences, Kunming 650223, China

**Keywords:** pore-forming toxin, red-belly toad, *Bombina maxima*, hippocampal neuronal cell, pyroptosis, cognitive function

## Abstract

Toxin-like proteins and peptides of skin secretions from amphibians play important physiological and pathological roles in amphibians. βγ-CAT is a Chinese red-belly toad-derived pore-forming toxin-like protein complex that consists of aerolysin domain, crystalline domain, and trefoil factor domain and induces various toxic effects via its membrane perforation process, including membrane binding, oligomerization, and endocytosis. Here, we observed the death of mouse hippocampal neuronal cells induced by βγ-CAT at a concentration of 5 nM. Subsequent studies showed that the death of hippocampal neuronal cells was accompanied by the activation of Gasdermin E and caspase-1, suggesting that βγ-CAT induces the pyroptosis of hippocampal neuronal cells. Further molecular mechanism studies revealed that the pyroptosis induced by βγ-CAT is dependent on the oligomerization and endocytosis of βγ-CAT. It is well known that the damage of hippocampal neuronal cells leads to the cognitive attenuation of animals. The impaired cognitive ability of mice was observed after intraperitoneal injection with 10 μg/kg βγ-CAT in a water maze assay. Taken together, these findings reveal a previously unknown toxicological function of a vertebrate-derived pore-forming toxin-like protein in the nerve system, which triggers the pyroptosis of hippocampal neuronal cells, ultimately leading to hippocampal cognitive attenuation.

## 1. Introduction

Pore-forming toxins (PFTs) are important exotoxins which are usually secreted by pathogenic bacteria and have a toxic function, mainly by perforating the target cell membrane and causing cell death [1]. Aerolysin is a β-barrel-type PFT primarily produced by Aeromonas species. Interestingly, an increasing number of genomic data and bioinformatic analyses have shown that aerolysin-like proteins (ALPs) widely exist in all living things, including vertebrates [2,3]. It is well known that pathogenic bacteria-derived ALPs exert various toxic effects (such as hemolysis, cell death, and others) via channel formation in a target cell’s membrane [4]. Regrettably, the detailed pathophysiologic functions of vertebrate-derived ALPs are mostly unknown; one of the most important reasons for this is that naturally purified ALPs are difficult to obtain [5]. Recently, emerging evidence has suggested that recombinant vertebrate-derived ALPs play pivotal toxicological and pathophysiological roles in living organisms. The gene encoding natterin-like ALPs from the *Danio rerio* fish has been found to play important physiological roles in the embryonic development of zebrafish [6,7]. Moreover, the recombinant ALP of *Danio rerio*, named Aep1, has also been found to play a crucial role in the antimicrobial immunity of zebrafish [8]. It has been reported that an ALP from lamprey, named LIP, can selectively kill tumor cells [9]. All of these findings indicate that ALPs from vertebrates are involved in many pathophysiologic processes, while the detailed biological functions and molecular mechanisms of natural ALPs remain unclear.

The hippocampal region of the brain is responsible for learning and memory, and the integrity and plasticity of hippocampal neurons are crucial to the maintenance of the learning and cognitive function of the hippocampus [10,11]. Damage to hippocampal neurons will obviously affect the learning and cognitive abilities of animals. A number of exotoxins secreted by bacteria have been reported to produce significant toxicity to neurons; one of the most representative types is neurotoxins [12,13], including botulinum neurotoxins, tetanus neurotoxins, cholera toxins, etc. Both botulinum neurotoxins and tetanus neurotoxins cause severe neuroparalytic syndromes dependent on their metalloprotease activities [14]. As the main member of the bacterial exotoxin family, PFTs are also toxic to the nervous system. The epsilon toxin (ETX), an ALP produced by *Clostridium perfringens*, has been identified to induce perivascular oedema of the brain and lead to the firing of the neural network by binding to certain neurons or oligodendrocytes [15]. However, the roles of vertebrate-derived ALPs in the nervous system are largely unclear, and to date, the effect of ALPs from vertebrates on the hippocampus has not been reported.

In our previous studies, an ALP complex composed of one ALP subunit and two trefoil factor (TFF) subunits, named βγ-CAT, was identified from the skin secretions of *Bombina maxima*, a red-belly toad distributed specifically in southwest China [16]. The results of previous studies revealed that βγ-CAT has diverse toxic effects via pore formation in the membrane of target cells, especially mammalian cells. βγ-CAT not only strongly induces the hemolysis of red blood cells but also triggers the Ca^2+^-dependent apoptosis of platelets [17]. Not surprisingly, an increasing amount of evidence has shown that βγ-CAT also displays diverse pharmacological activities in addition to typical toxic effects. Further studies showed that βγ-CAT can selectively kill tumor cells, promote wound healing and tissue repair, and counteract enveloped virus invasion by directly killing enveloped virus [18,19]. The study of detailed molecular mechanisms revealed that acidic glycosphingolipids in target cell membranes are the receptors of βγ-CAT and mediate the membrane binding, oligomerization, and endocytosis of βγ-CAT [20,21]. Acidic glycosphingolipids are mainly distributed in the nervous system, especially in the myelin sheath of neurons [22,23]. Unfortunately, the toxic roles of βγ-CAT in the nervous system remain elusive.

Here, the detailed role of βγ-CAT in the brain was explored, and our findings showed that βγ-CAT can cross the blood–brain barrier and induce the cell death of mouse hippocampal neuronal cells. Subsequent studies indicated that βγ-CAT-induced cell death was gasdermin-E- and caspase-1-dependent, so the βγ-CAT-induced cell death of mouse hippocampal neuronal cells is pyroptosis. Further studies revealed that the pyroptosis of mouse hippocampal neuronal cells induced by βγ-CAT relied on the membrane binding, oligomerization, and endocytosis of βγ-CAT. Finally, the cognitive ability of an animal was shown to be impaired after being treated with βγ-CAT. These findings provide previously unknown data which enable better understanding of the roles of vertebrate ALPs in the nervous system.

## 2. Results

### 2.1. βγ-CAT Displays Strong Cytotoxicity and Induces Death of Mouse Hippocapmal Neuronal Cells

It is well known that acidic glycosphingolipids (mainly gangliosides and sulfatides) are mostly distributed in neurons or glial cells [24]. Acidic glycosphingolipids in cell membrane are the receptors of βγ-CAT and are involved in the endocytosis of βγ-CAT, as described previously [20]. Thus, the cytotoxic effect of βγ-CAT on neurons was studied first. To test the effect of the cytotoxicity of βγ-CAT on neurons, the lactic dehydrogenase (LDH) release assay was performed, and the HT-22 mouse hippocampal neuronal cell line was used. The findings showed that the LDH release of HT-22 cells induced by βγ-CAT was largely increased at a concentration no lower than 5 nM, and the LDH release of HT-22 cells induced by βγ-CAT was shown to occur in a dose-dependent manner (Figure 1A). In addition, an MTT assay was also performed to determine the viability of HT-22 cells after being treated with βγ-CAT, and the findings showed that cell viability was significantly reduced after treatment with βγ-CAT at a starting concentration of 5 nM (Figure 1B). These findings suggest that βγ-CAT is highly cytotoxic to hippocampal neuronal cells and impairs cell survival. To further study HT-22 cell death induced by βγ-CAT, flow cytometry was performed. HT-22 cells underwent progressive cell death after being treated by 5 nM βγ-CAT at different treatment times, and this was shown in a time-dependent manner (Figure 2A). The further quantitative analysis of the dead cells revealed that significant cell death occurred after HT-22 cells were treated with 5 nM βγ-CAT for 3 min or more (Figure 2B). Taken together, these results show that βγ-CAT presented strong cytotoxicity to mouse hippocampal neuronal cells and further led to cell death. However, the type of cell death of hippocampal neuronal cells that βγ-CAT induces still requires further study.

### 2.2. βγ-CAT Triggers Gasdermin-E-Dependent Pyroptosis of HT-22 Cells

It is well known that, to date, many types of cell death have been identified, such as apoptosis, programmed cell necrosis, ferroptosis, etc. [25,26]. Pyroptosis is a type of inflammatory cell death triggered by excessive inflammation, which is generally considered to be accompanied by the activation of inflammasomes and the mature cleavage of gasdermins. Thus, pyroptosis is also defined as gasdermin-mediated programmed necrotic cell death [27]. To study the type of cell death induced by βγ-CAT, the transcriptomic analysis of HT-22 cells after treatment with βγ-CAT revealed that the expression of gasdermin E is upregulated after being treated with βγ-CAT, suggesting that the βγ-CAT-triggered cell death of HT-22 cells is pyroptosis. However, more evidence is still needed to support our hypothesis. In order to determine the type of gasdermin expressed on HT-22 cells, a polymerase chain reaction (PCR) and real-time quantification PCR (RT-qPCR) were performed, and the findings showed that gasdermin E (GSDME) was the predominantly expressed gasdermin on HT-22 cells (Figure 3A,B). A subsequent Western blotting assay revealed that both GSDMD and GSDME are expressed in Raw 264.7 cells, while only GSDME is expressed in HT-22 cells (Figure 3C). Furthermore, the expression of a cleaved GSDME N-terminal in HT-22 cells was increased significantly after treatment with βγ-CAT at a concentration of 3 nM for different lengths of time, and the expression of a cleaved GSDME N-terminal was presented in a time-dependent manner. In addition to GSDME, the expression levels of cleaved caspase 1, IL-1β, and IL-18 were also significantly increased in HT-22 cells after βγ-CAT treatment (Figure 3D). As mentioned above, the occurrence of pyroptosis is usually accompanied by the cleavage of gasdermin and the maturation and release of IL-1β or IL-18. A subsequent enzyme-linked immunosorbent assay (ELISA) showed that both IL-1β and IL-18 were largely increased in the cell supernatant after being treated with 3 nM βγ-CAT (Figure 3E,F). These findings showed that the cell death of HT-22 cells induced by βγ-CAT was GSDME- and caspase-1-dependent, suggesting that βγ-CAT has the ability to trigger the GSDME- and caspase-1-dependent pyroptosis of HT-22 cells.

### 2.3. The Pyroptosis of HT-22 Cells Triggered by βγ-CAT Depend on Its Membrane Binding, Oligomerization and Endocytosis

A previous study showed that βγ-CAT functions through a general mode of action, namely membrane binding, oligomerization, and endocytosis [20]. Here, the pyroptosis of HT-22 cells triggered by βγ-CAT was found like in the studies mentioned above, but the detailed molecular mechanisms remain unclear. First, flow cytometry analysis revealed that βγ-CAT binds strongly with the plasma membranes of HT-22 cells at the concentration of 2.5 or 5 nM (Figure 4A). In general, oligomerization and pore formation will occur once ALPs bind to the cell membrane. Therefore, Western blotting was used to detect the oligomerization of βγ-CAT. A sodium dodecyl sulfate (SDS)-stable oligomer at approximately 180 kDa could be observed after HT-22 cells were treated with βγ-CAT at the concentrations of 5 or 10 nM (Figure 4B), suggesting that βγ-CAT not only undergoes membrane binding but also undergoes allosteric and oligomerization processes. A previous study revealed that the endocytosis of βγ-CAT is pivotal for the functions of βγ-CAT, so βγ-CAT was determined to be a secretory endolysosome channel [28]. Thus, the endocytosis of βγ-CAT was further studied using confocal microscopy. The co-localization of βγ-CAT and lamp-1 could be observed after HT-22 cells were treated with 5 nM βγ-CAT (Figure 4C). The further quantitative analysis of the co-localization region revealed that the Pearson’s coefficient of the βγ-CAT treatment group was much higher than that of the control group (Figure 4D), suggesting that after endocytosis, βγ-CAT functions by regulating the endocytolysosomal pathway. Taken together, these findings suggest that the pyroptosis induced by βγ-CAT relies on the membrane binding, oligomerization, and endocytosis of βγ-CAT.

### 2.4. βγ-CAT Crosses the Blood-Brain Barrier and Reduces Cognitive Function of Mice

As mentioned above, βγ-CAT triggers the GSDME-dependent pyroptosis of mouse hippocampal neuronal cells through its membrane binding, oligomerization, and endo-cytosis. It is well known that the excessive death of hippocampal neuronal cells impairs cognitive function in the hippocampus [29,30]. Therefore, it is necessary to explore the role of βγ-CAT on the cognitive function of mice. It has been reported that many toxin-like proteins cannot cross the blood–brain barrier (BBB) because of their high molecular weight. βγ-CAT is a 72 kDa protein complex, and theoretically, it crosses the BBB with difficulty. Here, the ability of βγ-CAT to cross BBB was determined using an in vitro BBB model built with a co-culture of the human cerebral microvascular endothelial cell line (hCMEC/D3) and human normal glial cell line (HEB). Our findings revealed that the transendothelial electrical resistance (TEER) reduction was increased significantly after treatment with βγ-CAT at the concentrations of 2.5 or 5 nM, and the peak of TEER reduction was present 1 h after βγ-CAT treatment (Figure 5A). Similarly, the penetration rate of Na–F was also increased after βγ-CAT treatment (Figure 5B). To further test the penetrability of βγ-CAT on the BBB, an in vivo BBB penetration assay was performed using the fluorescent tracer method. The whole brain of a mouse was removed two hours after the mice were intraperitoneally injected with 100 μL of 40 μg/mL βγ-CAT. The immunofluorescence analysis revealed that βγ-CAT crosses the BBB and is enriched in the hippocampus (Figure 5C). These findings suggest that βγ-CAT treatment augments the permeability of the BBB and then crosses the BBB. To further study the impaired hippocampal synaptic functions and spatial cognitive functions induced by βγ-CAT, a water maze assay was adopted. The findings from the water maze assay revealed that the escape latency of mice during the spatial learning stage increased significantly after intraperitoneal injection with 10 μg/kg βγ-CAT (Figure 6A). In addition to the escape latency, the dwell time during the memory test also increased after intraperitoneal injection with 10 μg/kg βγ-CAT (Figure 6B). In addition to time, the distance in the water maze was also measured, and the findings showed that the moving distance in the center also increased after intraperitoneal injection with 10 μg/kg βγ-CAT (Figure 6C). Curiously, the percentage time in the center did not significantly increase after intraperitoneal injection with 10 μg/kg βγ-CAT (Figure 6D). These findings reveal that βγ-CAT has the ability to pass through the BBB and then trigger the pyroptosis of hippocampal neuronal cells; finally, it impairs the cognitive functions of the hippocampus.

## 3. Discussion

Toxins are not only important weapons/arsenals to enable poisonous animals to hunt prey/defend themselves, but they are also efficient and active pharmacological molecules that can be used by humans [5]. In addition, the regulation of toxins on the physiological processes of organisms is also an important research field. βγ-CAT is a pore-forming toxin-like protein complex which was first identified in *Bombina maxima*, a Chinese red-belly toad. Previous studies revealed that βγ-CAT has a variety of physiological regulatory functions (such as immune regulation, wound healing, tissue repair, etc.). A further molecular mechanism study indicated that βγ-CAT functions predominantly by regulating the endolysosome pathway and the general action mode of βγ-CAT functions, including membrane binding, oligomerization, and endocytosis. Based on this action mode of βγ-CAT, the roles of βγ-CAT in the nervous system were further studied, and the findings showed that βγ-CAT can trigger the gasdermin-E-dependent pyroptosis of hippocampal neuronal cells and impairs the cognitive functions of the hippocampus.

The effects of biotoxins on the nervous system have been reported previously; for example, α-bungarotoxin, a neurotoxin derived from the venom of the *Bungarus mul-ticinctus* snake, has been found to block neuromuscular transmission by binding the nicotinic acetylcholine receptor α-subunit in skeletal muscles [31]. The cholera toxins produced by *Vibrio cholerae* affect the normal neural functions of neurons and glial cells by interacting with ganglioside GM1 in neurons in the cerebral cortex [32]. As the main family of pore-forming toxins, bacterial-secreted ALPs have various pathophysiological roles in the nervous system [33]. One of the representative ALPs is the epsilon toxin (ETX), which is mainly produced by *Clostridium perfringens* and has been found to be a cause of central nervous system white matter disease in ruminant animals [34]. However, the detailed biological functions of vertebrate-derived ALPs remain elusive. Particularly, the roles of vertebrate-derived ALPs in the nervous system remain unclear to date. There are two primary reasons for this: (1) natural and active ALPs are difficult to obtain from vertebrates and (2) the huge molecular weights of most ALPs mean they are less likely to be able to cross the BBB. Indeed, the existence of the BBB is crucial to the protection of the function of the brain, and many macromolecular exogenous substances are isolated from the BBB because of their antigenicity and heterogeneity [35]. However, there is no absolute relationship between molecular weight and BBB penetration [36]. A recent study revealed that the epsilon toxin (ETX) can cross the BBB by binding to the lymphocyte protein (MAL) on the luminal surface of endothelial cells [37]. Thus, macromolecules with high molecular weights are also likely to cross the BBB in a unique way. Coincidentally, βγ-CAT is a toxin-like protein with a high molecular weight of 72 kDa, so theoretically, βγ-CAT penetrates the BBB with difficulty. Surprisingly, in the present study, first-hand evidence was provided which supported the concept that βγ-CAT can cross the BBB (Figure 5). This can be explained from the perspective of the receptor of βγ-CAT. Acidic glycosphingolipids in cell membranes are the receptors of βγ-CAT and are involved in the membrane binding, oligomerization, and endocytosis of βγ-CAT, as described previously [20]. In fact, acidic glycosphingolipids (mainly gangliosides and sulfatides in vertebrates) are predominantly distributed in the nervous system, especially in the brain [38]. In addition, gangliosides and sulfatides are also highly expressed by endothelial cells and astrocytes, which make up the BBB [39,40], which enables βγ-CAT to easily bind to the BBB and then cross it. Why is βγ-CAT specifically localized in the hippocampus after crossing the BBB and entering the brain? One possible reason for this is that besides acidic glycosphingolipids, an unknown protein receptor of βγ-CAT may specifically exist in the hippocampus. However, more detailed molecular mechanisms need to be further studied.

The central nervous system currently consists of three major cell types: neurons, astrocytes, and microglia [41,42]. Currently, microglia are reported to be mainly involved in immune function; the excessive release of pro-inflammatory factors and gasdermin activation in microglia will lead to the pyroptosis of the whole nervous system [43,44]. In the present study, evidence showed that only GSDME is expressed in hippocampal neuronal cells, not GSDMD. There have been controversial reports regarding the expression patterns of gasdermin families in HT-22 cells. Some scholars believe that HT-22 cells are expressed in GSDMD [45], while others believe that HT-22 cells are exclusively expressed in GSDME [46], while other gasdermin family members are not. In our study, both RT-qPCR and Western blotting showed that HT-22 cells express GSDME, rather than GSDMD (Figure 3A–C). Meanwhile, we further confirmed that βγ-CAT induced the pyroptosis of HT-22 cells in a GSDME-dependent manner (Figure 3D–F). Our findings provided convincing first-hand evidence of the involvement of hippocampal neurons in inflammatory process.

Some physiological pore-forming toxin-like proteins or peptides in the human body are also involved in the development of various neurodegenerative diseases under pathological conditions [47,48]. Amyloid β peptide (Aβ), a peptide with membrane-perforating activity which has been identified in the human brain, was found to be involved in the pathogenesis of Alzheimer’s disease [49,50]. Detailed studies revealed that cholesterol and gangliosides in cell membranes are potential binding sites of Aβ [51,52,53]. The pore-forming characteristics and mechanisms of Aβ are obviously similar to those of βγ-CAT reported in this study. Unfortunately, the detailed pathogenic mechanisms of Aβ in Alzheimer’s disease or Parkinson’s disease remain elusive [54]. Therefore, βγ-CAT is expected to be an appropriate candidate for the exploration of the detailed pathogenic mechanism of Aβ in Alzheimer’s disease or other neurodegenerative diseases.

The primary function of the toxins secreted by the skin of amphibians is defensive, rather than predatory [55,56]. However, the detailed roles of βγ-CAT in defense against predators remain elusive. Here, our findings showed that βγ-CAT impaired the cognitive memory functions of animals by triggering the GSDME-dependent pyroptosis of hippocampal neurons (Figure 6). Based on these findings, we hypothesize that after a predator preys on the red-belly toad for the first time, a symptom of poisoning, mainly entailing “decreased learning and memory function”, will occur in the predator, meaning the predator will not be able to remember its prey clearly. Eventually, this will prevent the toad from being eaten by its predators. Perhaps this is the main reason for the large amount of βγ-CAT which exists in the frog’s skin secretions.

In summary, in our study, we systematically explored the detailed toxic functions and mechanisms of βγ-CAT in hippocampal neuronal cells and the hippocampus of the brain, further enriching our knowledge of the toxic effects and physiological functions of this toxin. Importantly, this work will lay a theoretical foundation for the subsequent study of the biological functions of ALPs in vertebrates.

## 4. Conclusions

The toxic effects and biological functions of vertebrate-derived ALPs remain unclear. βγ-CAT was the first natural ALP and TFF complex to be identified in vertebrates. Previous studies revealed that βγ-CAT exerts various pathophysiological roles via its regulation of the endolysosome pathway. A further study showed that acidic glycosphingolipids are the receptors of βγ-CAT and mediate the membrane binding, oligomerization, and endocytosis of βγ-CAT. As acidic glycosphingolipids are mainly distributed in the nervous system, the functions of βγ-CAT in the nervous system were explored in the present study. The main conclusions from the study are listed below:(1)βγ-CAT shows strong cytotoxicity to HT-22 hippocampal neuronal cells and triggers the GSDME-dependent pyroptosis of HT-22 cells.(2)The pyroptosis of HT-22 cells induced by βγ-CAT relies on the membrane binding, oligomerization, and endocytosis of βγ-CAT.(3)βγ-CAT can cross the BBB and affects the cognitive function of mice.

## 5. Materials and Methods

### 5.1. Animals

The ICR mice weighing 18 to 22 g were used in the study and purchased from Hunan SJA Laboratory Animal Co., Ltd. (Changsha, China). The mice were housed in IVC cages under constant temperature (23 ± 1 °C) and humidity with a 12 h light/dark cycle. All procedures and the care and handling of animals were approved by the Ethics Committee of Yunnan Normal University (No.YSLL20220316).

### 5.2. Cell Lines, Antibodies and Reagents

The mouse hippocampal neuronal cell line HT-22 was purchased from the American Type Culture Collection (Manassas, VA, USA) and maintained in Dulbecco’s modified Eagle’s medium (DMEM) supplemented with 10% fetal calf serum (FCS). The mouse macrophage Raw 264.7 cells were cultured in DMEM medium supplemented with 10% FCS. Natural βγ-CAT was purified from the skin secretions of *Bombina maxima* as described previously [16]. The LDH cytotoxicity assay kit was purchased from Beyotime Biotechnology Inc. (Shanghai, China). The MTT assay kit (ab211091) was purchased from Abcam Inc. (Cambridge, CB2 0AX, UK). The Annexin V-FITC apoptosis detection kit (APOAF-50TST) was purchased from Sigma-Aldrich (St. Louis, MO, USA). The anti-GSDME (ab215191), anti-Lamp1 antibody (ab25630), anti-GSDMD (ab219800), and anti-IL-18 (ab191860) antibodies were purchased from Abcam Inc. (Cambridge, CB2 0AX, UK). The IL-1β (3A6) mouse monoclonal antibody (12242S) was purchased from Cell Signaling Technology, Inc. (Danvers, MA, USA). The anti β-actin antibody (81115-1-RR) was purchased from Proteintech Group, Inc. (Wuhan, Hubei, China). Mouse IL-18 ELISA kit (SEKM-0019) and IL-1β ELISA kit (SEKM-0002) were purchased from Solarbio Life Sciences (Beijing, China). Anti-βγ-CAT antibody was prepared by our lab. The caspase 1 p20 antibody (sc-398715) and GAPDH antibody (sc-365062) were purchased from Santa Cruz Biotechnology (Santa Cruz, CA, USA). The Cy3-conjugated goat anti-mouse IgG/IgM (H+L) secondary antibody (M30010) and the FITC-conjugated goat anti-rabbit IgG/IgM (H+L) secondary antibody (F-2765) were purchased from Thermo Fisher Scientific Inc. (Waltham, MA, USA). All other reagents were purchased from Sigma-Aldrich (St. Louis, MO, USA).

### 5.3. LDH Release Detection

The LDH release measurement of HT-22 cells after treated by βγ-CAT was used to assess the cytotoxicity of βγ-CAT to mouse hippocampal neuronal cells. The measurement procedure of LDH release in this study was performed as previously described [20]. First, the HT-22 cells were cultured in DMEM medium supplemented with 10% FCS and the cells were harvested and seeded into a 96-well plate when the cells grew to 80% confluence. After the cells grew to 80–90% confluence, the medium with 10% FCS was replaced by serum-free medium and then cultured for 2 h. Next, the serum-free medium of each well was removed and washed three times with cold phosphate buffer saline (PBS), then 100 μL βγ-CAT with different concentrations was added to each well and incubated at 37 °C for 10 min. At last, the cell supernatant of each well was collected and the LDH content in cell supernatant was detected according to the manufacturer’s instructions. The HT-22 cells treated with 0.1% Triton X-100 was known as 100% LDH release, and the cells treated with PBS served as a negative control.

### 5.4. MTT Assay

To detect the cell viability of HT-22 cells after treated by βγ-CAT, the MTT assay was performed. In brief, the HT-22 cells were harvested and planted into a 96-well plate. After the cells grew to 80–90% confluence, the medium with 10% FCS was replaced by serum-free medium and then cultured for 2 h. Next, the cells were incubated with gradient concentrations of βγ-CAT (from 2.5 to 20 nM) at 37 °C in water bath for 10 min. After incubation, the βγ-CAT of each well was removed and washed three times with cold PBS. Then, 120 μL MTS reagents were added to each well and incubated at 37 °C for 2 h. Finally, the absortance at 490 nm of each well was recorded by using a 96-well plate reader. Cells only treated with βγ-CAT served as a positive control and the cells treated with FBS-free medium served as a negative control. The HT-22 cells treated with PBS were known as 100% survival and cells treated by 0.1% Triton X-100 were known as 0% survival.

### 5.5. Flow Cytometry Detection of HT-22 Cell Death

The method of flow cytometry was used to detect the cell death of HT-22 cells after being treated by βγ-CAT as previously described [57]. Briefly, the HT-22 cells were harvested when the cells grew to 80–90% confluence, and then the cells were washed three times with cold PBS. The washed cells were next treated with 5 nM βγ-CAT at 37 °C for gradient times (from 3 to 10 min), and then the HT-22 cells were collected and washed three times with cold PBS. The washed cells were then incubated with FITC labeled Annexin V and propidium iodide (PI). The mixture of cells and FITC Annexin V with PI was vortexed gently and incubated at 25 °C for 15 min in the dark. Finally, the cell supernatant was collected and analyzed by flow cytometry. The unstained cells served as a blank control. The cells only stained with FITC Annexin V or only stained with PI also served as the control. The HT-22 cells untreated with βγ-CAT were served as a negative control.

### 5.6. PCR and RT-qPCR Assay

The methods of PCR and RT-qPCR were employed to detect the expression of gasdermins in HT-22 cells as previously described [58]. Briefly, the cultured HT-22 cells were collected by digesting with trypsin. Then, the total RNA was extracted by using the TRIzol reagent according to the manufacturer’s instructions. Next, the cDNA first-strand was synthesized by using a reverse transcription system according to the manufacturer’s instructions. To detect the expression profile of gasdermins in HT-22 cells, the specific gasdermin primers for PCR and RT-qPCR were designed using Oligo 7 software and listed in Table 1. It is worth mentioning that the GSDMB primer designed here was based on human GSDMB isoform since no remarkable GSDMB sequence can be found in GenBank. In PCR assay, 35 cycles were performed and amplified by using taq DNA polymerase. The RT-qPCR assay was performed by using the SYBR Premix Ex Taq II two-step qRT-PCR kit according to the manufacturer’s instructions on a LightCycler 480 real-time PCR system (Roche LightCycler 480, Roche, Mannheim, Germany). The relative expression of gasdermins was determined by CT value analysis of reference genes and target genes by Pfaffl method.

### 5.7. Western Blotting Assay

Firstly, the Western blotting assay was adopted to detect the expression of GSDMD or GSDME in HT-22 cells. Briefly, the cultured HT-22 cells were collected by digesting with trypsin and lysed with cell lysis buffer containing protease inhibitor cocktail. The total proteins of HT-22 cells were extracted and quantified by BCA method. Then, the total protein sample was separated by 12% sodium dodecyl sulfate-polyacrylamide gel electrophoresis (SDS-PAGE) and then electrotransferred onto polyvinylidene difluoride (PVDF) membranes. The PVDF membrane was subsequently blocked with 5% skimmed milk containing 0.1% Tween-20 at room temperature for 2 h. Then, the membrane was incubated with rabbit anti-GSDMD and rabbit anti-GSDME antibodies (1:1000 diluted) at 4 °C overnight. Next, the primary antibodies were removed and then incubated with HRP-conjugated goat anti-rabbit secondary antibodies (1:5000 diluted). At last, the protein bands were visualized with the SuperSignal WestPico chemiluminescence substrate via Gel Imager System.

To detect the expression of GSDME, cleaved GSDME N-terminal (NT), caspase 1, cleaved caspase 1 p20, IL-1β, and IL-18 of HT-22 cells, the HT-22 cells were treated by 5 nM βγ-CAT in a time gradient from 2 h to 8 h, and then washed three times with cold PBS to remove the βγ-CAT. The collected HT-22 cells were then lysed to obtain the total proteins. Then, the total protein sample was separated by 12% sodium dodecyl sulfate-polyacrylamide gel electrophoresis (SDS-PAGE) and electrotransferred onto polyvinylidene difluoride (PVDF) membranes. The PVDF membrane was subsequently blocked with 5% skimmed milk containing 0.1% Tween-20 at room temperature for 2 h. Then, the membrane was incubated with rabbit anti-GSDMD antibody (1:1000 diluted), mouse anti-caspase-1 p20 antibody (1:500 diluted), mouse anti-IL-1β monoclonal antibody (1:1000 diluted), rabbit anti-IL-18 antibody (1:1000 diluted), and rabbit anti-β actin antibody (1:5000 diluted) at 4 °C overnight. Next, the primary antibodies were removed and then incubated with HRP-conjugated goat anti-rabbit secondary antibody and HRP-conjugated goat anti-mouse secondary antibody (1:5000 diluted). At last, the protein bands were visualized with the SuperSignal WestPico chemiluminescence substrate via Gel Imager System.

To detect the oligomerization of βγ-CAT on the membrane of HT-22 cells, the HT-22 cells were treated by 5 nM and 10 nM βγ-CAT at 37 °C for 30 min, in which the treated cells were collected and lysed. After SDS-PAGE and transfer to PVDF, the PVDF membrane was subsequently blocked with 5% skimmed milk containing 0.1% Tween-20 at room temperature for 2 h. Then, the membrane was incubated with rabbit anti-βγ-CAT antibody (1:1000 diluted) and HRP-conjugated goat anti-rabbit secondary antibody (1:5000 diluted). Finally, the protein bands were visualized with the SuperSignal WestPico chemiluminescence substrate via Gel Imager System.

### 5.8. Enzyme-Linked Immuno Sorbent Assay (ELISA)

To determine the IL-1β and IL-18 levels in the cell supernatant of HT-22 cells after being treated with βγ-CAT, the ELISA was performed. Briefly, the HT-22 cells were treated by 5 nM βγ-CAT at a time gradient from 2 h to 8 h, and then the cell supernatant was collected and the IL-1β and IL-18 of the supernatant were measured by the commercial ELISA kit according to the manufacturer’s instructions.

### 5.9. Flow Cytometry Detection of the Membrane Binding of βγ-CAT

The method of flow cytometry was used to detect the membrane binding of βγ-CAT with HT-22 cells. Briefly, the HT-22 cells were treated by βγ-CAT at the concentrations of 2.5 and 5 nM for 15 min, and then βγ-CAT was removed and the cells were washed three times with cold PBS. The cells were incubated with rabbit anti-βγ-CAT antibody (1:500 diluted) and FITC-conjugated goat anti-rabbit secondary antibody (1:500 diluted). Finally, the cells were resuspended in 300 μL of PBS and analyzed on a flow cytometer (FACSVantage SE; Becton Dickinson, Franklin Lakes, NJ, USA). Data were analyzed using FlowJo software 7.6.1 (Tree Star Inc., Ashland, OH, USA)

### 5.10. Confocal Microscopy Assay of the Endocytosis and Co-Localization of βγ-CAT

To detect the endocytosis of βγ-CAT and the co-localization between βγ-CAT and endolysosome, the confocal microscopy assay was performed as previously described [20]. In brief, the HT-22 cells were treated by βγ-CAT at the concentrations of 5 nM for 30 min, and then incubated with rabbit anti-βγ-CAT polyclonal antibody (1:500 diluted) and mouse anti-lamp-1 antibody (1:200 diluted). After being washed three times, the HT-22 cells were then incubated with FITC-conjugated goat anti-rabbit secondary antibody (1:500 diluted) and Cy3-conjugated goat anti-mouse secondary antibody (1:500 diluted). Finally, the nuclei were stained with DAPI. After being washed three times, the slides were observed under a confocal microscope (Olympus FV1000, Olympus Corporation, Tokyo, Japan). For the co-localization analyses of βγ-CAT with Lamp-1, the region of interests in HT-22 cells were analyzed using the “Just Another Co-localization Plugin (JACoP)” of Image J. The offset of each image was set automatically to avoid an arbitrary judgement, and then the Pearson’s correlation coefficient was calculated as previously described [59].

### 5.11. In Vitro BBB Model Establishment and βγ-CAT Penetration Detection

To detect the ability of βγ-CAT to penetrate the BBB, an in vitro BBB model was built with a co-culture of the human cerebral microvascular endothelial cell line (hCMEC/D3) and the human normal glial cell line (HEB), as previously described [60,61]. Briefly, the cultured HEB cells at cell numbers of 5 × 10^4^ cells/cm^2^ were first seeded on the bottom of the 24-well Transwell inserts for 24 h until they had adhered. Subsequently, the hCMEC/D3 cells at cell numbers of 5 × 10^5^ cells/cm^2^ were seeded on the top side of the Transwell chamber. After being co-cultured for 4–6 days, the integrity of the BBB model was determined via TEER measurement using Millicell ERS-2 (Millipore, MA, USA). The criterion for a successful in vitro BBB model was that the TEER value was higher than 200 Ω·cm^2^.

Based on the in vitro BBB model, the ability of βγ-CAT to penetrate the BBB was determined. Briefly, the in vitro BBB cell models (in the Transwell chamber) were treated with βγ-CAT at the concentrations of 2.5 and 5 nM for different lengths of time (0.5 h, 1.0 h, 1.5 h, 2.0 h, 2.5 h, and 3.0 h); then, the TEER values were recorded using Millicell ERS-2 (Millipore, MA, USA). In addition, sodium fluorescein (Na-F) was also used to test the paracellular permeability of the BBB. After βγ-CAT treatment, a 10 mg/L Na–F solution was then added to the in vitro BBB model. Finally, the concentration of Na–F was measured using a fluorescence multiwall plate reader. The untreated BBB model was used as a negative control.

### 5.12. In Vivo BBB Penetration Assay

To further study the BBB penetration activity of βγ-CAT, the in vivo assay was performed according to previously described methods [62]. Briefly, C57 mice were divided into 2 groups. Each mouse in the control group was intraperitoneally injected with 100 μL of normal saline, and each mouse in the experimental group was intraperitoneally injected with 100 μL of 40 μg/mL βγ-CAT. Two hours after the intraperitoneal injection, the mice were euthanized via CO_2_ inhalation, and the brain tissues were quickly extracted. The brain tissue was fixed with 4% paraformaldehyde and then embedded in paraffin and sectioned. The slices were then blocked with 5% goat serum, incubated with the rabbit anti-βγ-CAT primary antibody, incubated with 488-labeled goat anti-rabbit fluorescent secondary antibody, and finally, the fluorescence in the brain tissue was observed with a fluorescence microscope.

### 5.13. Water Maze Assay for Assessment of Spatial Learning and Memory of Mice

In our study, a water maze assay was performed to detect the cognitive functions (especially spatial learning and memory functions) of mice after intraperitoneal injection with 10 μg/kg βγ-CAT. The detailed procedure for the water maze assay was previously described [63,64]. A circular tank with a 150 cm diameter and 60 cm height served as the site of the water maze; then, the tank was divided into 4 equal quadrants (I, II, III, and IV). A platform with a 10 cm diameter was located at the center of quadrant III and 2 cm below the water surface. The water temperature in the tank was maintained at 24 ± 2 °C. The movement of the mice in the tank was monitored using a computerized video tracking system. It is well known that spatial learning is the first step in a water maze assay. In the period of spatial learning, the untreated and βγ-CAT-treated mice were required to undergo trials on each training day for 6 consecutive days. The mice were allowed to swim freely to find the platform in the tank. The time taken to find the platform (escape latency) was recorded. One day after spatial learning, memory training was performed. In this training procedure, the platform was taken out from the tank, and the mice were placed at the starting point of quadrant I. The quadrant dwell time was recorded during the 60 s memory test. Furthermore, an open field assay was also performed by using a square plastic arena. In this arena, mice were permitted to freely explore the arena for 20 min; then, the distance traveled and time spent in the central region were recorded.

### 5.14. Statistical Analysis

The data in this paper are presented as mean ± SD. The experiments were repeated independently at least two times. All data in the paper are representative of three independent experiments and were analyzed using GraphPad Prism 8.0 software. The significance of differences in LDH release assay and MTT assay was determined by unpaired two-tailed Student’s *t*-test. The comparative analysis of escape latency and dwell time in water maze assay were performed by using unpaired two-tailed Student’s *t*-test.

## Figures and Tables

**Figure 1 toxins-15-00191-f001:**
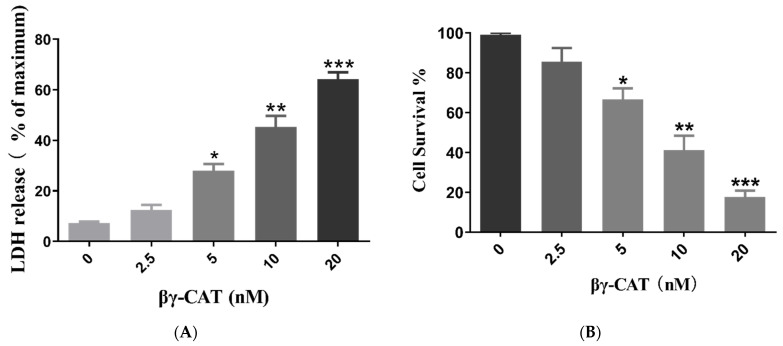
βγ-CAT has strong cytotoxicity to HT-22 mouse hippocampal neuronal cells. (**A**) HT-22 cells were treated with a gradient of concentrations of βγ-CAT (from 2.5 to 20 nM) at 37 °C for 10 min; then, the LDH release of HT-22 cells and the cytotoxicity of βγ-CAT was determined using LDH release assay. (**B**) HT-22 cells were treated with a gradient of concentrations of βγ-CAT (from 2.5 to 20 nM) at 37 °C for 10 min; then, the survival rate of HT-22 cells was measured using the MTS assay. * *p*  <  0.05, ** *p  *<  0.01, and *** *p*  <  0.001 versus the respective control. An unpaired two-tailed Student’s *t*-test was used in LDH release assay and MTT assay.

**Figure 2 toxins-15-00191-f002:**
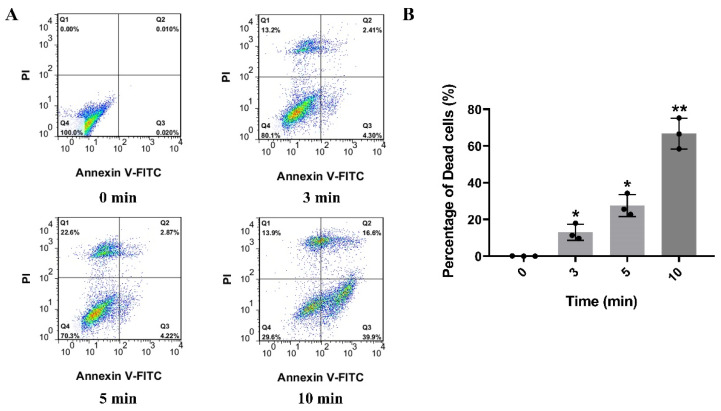
Flow cytometry analysis revealed that βγ-CAT can trigger cell death of HT-22 cells. (**A**) HT-22 cells were treated with 5 nM βγ-CAT at 37 °C with a gradient of times (from 3 to 10 min); then, the HT-22 cells were collected and washed three times and incubated with FITC-labeled Annexin V monoclonal antibody and PI. Finally, the cell death was determined via flow cytometry. (**B**) HT-22 cells were treated with 5 nM βγ-CAT at 37 °C with a gradient of times (from 3 to 10 min), and the percentage of dead cells was calculated and quantified. * *p*  <  0.05 and ** *p*  <  0.01 versus the control. An unpaired two-tailed Student’s *t*-test was used in flow cytometry assay. The data of flow cytometry are representative of three independent experiments.

**Figure 3 toxins-15-00191-f003:**
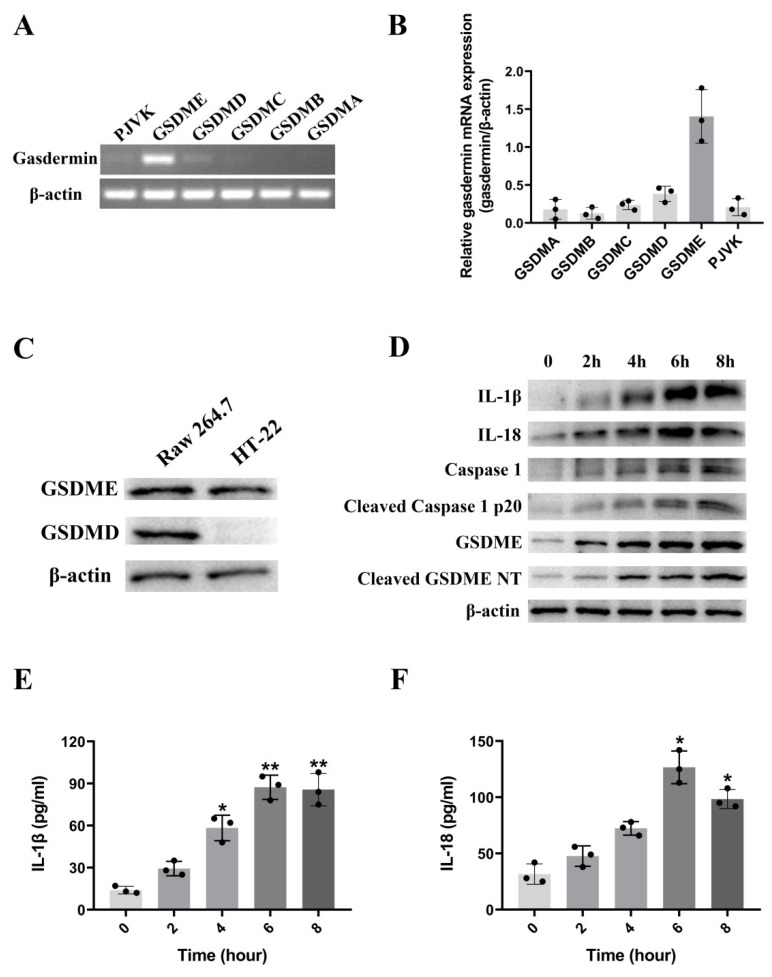
βγ-CAT triggers GSDME-dependent pyroptosis of HT-22 mouse hippocampal neuronal cells. (**A**) The expression of gasdermins in HT-22 cells was determined via PCR assay using specific primers. (**B**) The expression of gasdermins in HT-22 cells was determined via RT-qPCR assay using specific primers. (**C**) The expression of gasdermins in HT-22 cells was determined via Western blotting assay using specific antibodies of GSDMD and GSDME. (**D**) HT-22 cells were incubated with 3 nM βγ-CAT at 37 °C for different lengths of time (0 h, 2 h, 4 h, 6 h, and 8 h), and the full-length GSDME, cleaved GSDME N-terminal, caspase 1, cleaved caspase 1 p20, IL-1β, and IL-18 were detected via Western blotting using specific antibodies. (**E**) HT-22 cells were incubated with 3 nM βγ-CAT at 37 °C for different lengths of time (0 h, 2 h, 4 h, 6 h, and 8 h); then, the IL-1β in cell supernatant was measured via ELISA. * *p*  <  0.05 and ** *p*  <  0.01 versus the control. An unpaired two-tailed Student’s *t*-test was used in ELISA assay. (**F**) HT-22 cells were incubated with 3 nM βγ-CAT at 37 °C for different times (0 h, 2 h, 4 h, 6 h, and 8 h); then, the IL-18 in cell supernatant was measured via ELISA. * *p*  <  0.05 versus the control. An unpaired two-tailed Student’s *t*-test was used in ELISA assay. All immunoblots in (**C**,**D**) are representative of three independent experiments. The 3 dots on each column in (**B**,**E**,**F**) represent the data of 3 independent repeated experiments.

**Figure 4 toxins-15-00191-f004:**
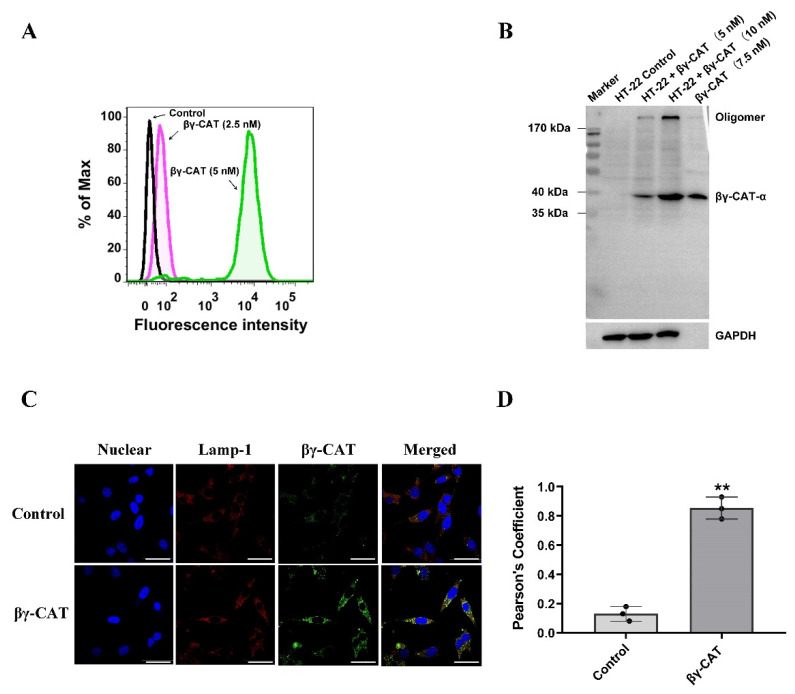
The pyroptosis triggered by βγ-CAT depended on the membrane, oligomerization, and endocytosis of βγ-CAT. (**A**) The HT-22 cells were treated with βγ-CAT at the concentrations of 2.5 and 5 nM for 15 min and then incubated with anti-βγ-CAT antibody; finally, the membrane binding of βγ-CAT was determined via flow cytometry. The untreated HT-22 cells were used as a negative control. (**B**) The HT-22 cells were treated with βγ-CAT at the concentrations of 5 and 10 nM for 30 min and then incubated with rabbit anti-βγ-CAT polyclonal antibody and goat anti-rabbit secondary antibody; finally, the oligomerization of βγ-CAT was determined via Western blotting. (**C**) The HT-22 cells were treated via βγ-CAT at the concentrations of 5 nM for 30 min and then incubated with rabbit anti-βγ-CAT polyclonal antibody and mouse anti-lamp-1 antibody. After being washed three times, the HT-22 cells were then incubated with FITC-labeled goat anti-rabbit secondary antibody and Cy3-labeled goat anti-mouse secondary antibody; finally, the endocytosis and lysosomal colocalization of βγ-CAT was determined via confocal microscopy. Scale bar = 25 μm. (**D**) Pearson’s correlation coefficients for co-localization of Lamp-1 and βγ-CAT. Data are expressed as mean ± SD of 4–6 cells. ** *p*  <  0.01 versus the control. All immunoblots in (**B**) are representative of three independent experiments.

**Figure 5 toxins-15-00191-f005:**
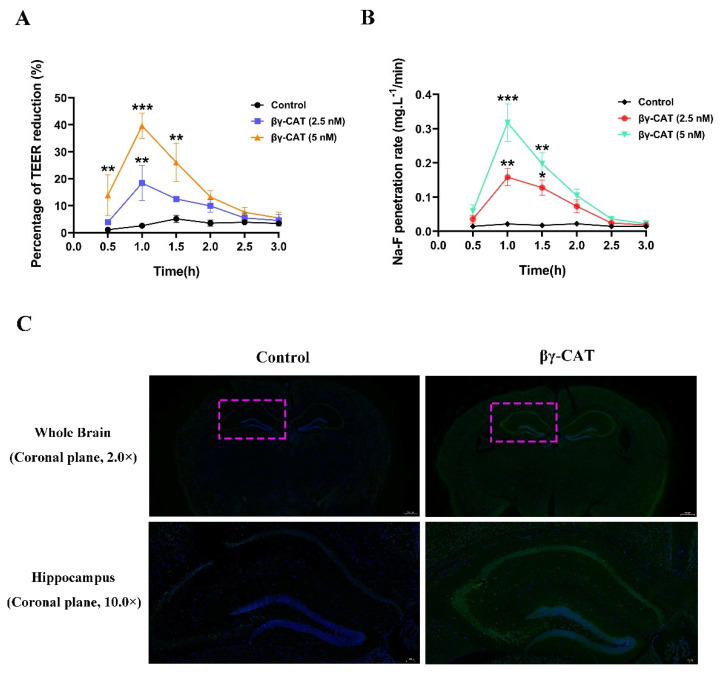
βγ-CAT crosses the BBB and is enriched in the hippocampus. It impairs the cognitive functions of mice. (**A**) The in vitro BBB cell models were treated with βγ-CAT at the concentrations of 2.5 and 5 nM for different lengths of time (0.5 h, 1.0 h, 1.5 h, 2.0 h, 2.5 h, and 3.0 h); then, the TEER values were determined with a Millicell ERS-2 instrument. The untreated BBB model was used as a negative control. * *p*  <  0.05, ** *p*  <  0.01 and *** *p*  <  0.001 versus the control. (**B**) The in vitro BBB cell models were treated with βγ-CAT at the concentrations of 2.5 and 5 nM for different lengths of time (0.5 h, 1.0 h, 1.5 h, 2.0 h, 2.5 h, and 3.0 h); then, the Na–F penetration rate was determined using a fluorescence multiwall plate reader. The untreated BBB model was used as a negative control. * *p*  <  0.05, ** *p*  <  0.01 and *** *p*  <  0.001 versus the control. (**C**) The mouse was intraperitoneally injected with 100 μL of 40 μg/mL βγ-CAT; then, the brain was extracted 2 h after intraperitoneal injection. The brain slices were fixed and incubated via rabbit anti-βγ-CAT antibody, then stained with 488-labeled goat anti-rabbit fluorescent secondary antibody, and finally, the fluorescence in the brain tissue was observed with a fluorescence microscope. The area marked by the magenta dotted frame is the hippocampus. Scale bar = 500 μm (upper) and 100 μm (lower).

**Figure 6 toxins-15-00191-f006:**
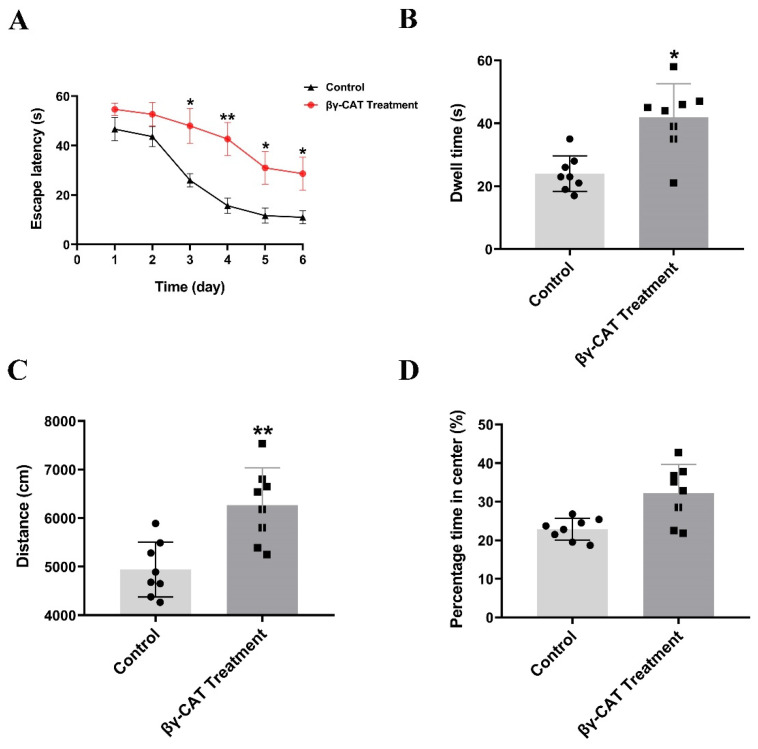
βγ-CAT impairs the hippocampal cognitive functions of mice. (**A**) The mice were intraperitoneally injected with 10 μg/kg βγ-CAT and then underwent a spatial learning test; the escape latency to reach the hidden platform was measured. The untreated mice were used as the negative control. * *p*  <  0.05 and ** *p*  <  0.01 versus control. (n = 8). (**B**) The memory test was conducted after the spatial learning test; the dwell time was recorded in a 60 s memory test. The untreated mice were used as the negative control. * *p*  <  0.05 versus control. (n = 8). (**C**) The open field test was conducted after the mice were intraperitoneally injected with 10 μg/kg βγ-CAT; the total travel distance was measured. The untreated mice were used as the negative control. ** *p*  <  0.01 versus control. (n = 8). (**D**) The open field test was conducted after the mice were intraperitoneally injected with 10 μg/kg βγ-CAT; the percentage of time spent in the center of the apparatus was measured. The untreated mice were used as the negative control (n = 8). The 8 dots on each column in (**B**–**D**) represent the data of 8 individual animals in each group in a representative experiment.

**Table 1 toxins-15-00191-t001:** Sequences of primers used for detection of gasdermins of HT-22 cell.

Primer	Sequence	Product Length
PCR		
GSDMA-Forward	TCCCTCCTGGAGAAAAGCCA	338 bp
GSDMA-Reverse	ACTTAGCACTGTCAGAGCCC	
GSDMB-Forward	TGGATGCCGGCACTACACAAC	248 bp
GSDMB-Reverse	GGTAGTTCCCTCTTCAGCTTCC	
GSDMC-Forward	GATCTGAGGCCTGTCAAATGC	524 bp
GSDMC-Reverse	TCTGTTTGCCACTGTCCACT	
GSDMD-Forward	CCGGAGTGTTTTGGCTCCTT	260 bp
GSDMD-Reverse	ACCACAAACAGGTCATCCCC	
GSDME-Forward	GTCAGCGCACTAGCAGAAATG	173 bp
GSDME-Reverse	ATGCCAAACCTCTCTGTGTC	
PJVK-Forward	GCTGACAAGTACCAACCCCT	595 bp
PJVK-Reverse	CACAAATGTCGAAGGCACCG	
β-actin-Forward	CCACCATGTACCCAGGCATT	253 bp
β-actin-Reverse	AGGGTGTAAAACGCAGCTCA	
RT-qPCR		
GSDMA-Forward	TCCCTCCTGGAGAAAAGCCA	261 bp
GSDMA-Reverse	GTGCTTCCAGGGTCACTTCG	
GSDMB-Forward	CCGTTAGAAGCCTTGTTGATGC	180 bp
GSDMB-Reverse	CCGTTGAGTCTACATTATCCAG	
GSDMC-Forward	CAGATGCAACCAAATTCTGCC	207 bp
GSDMC-Reverse	TGGTTTCGACATCCAGGTCA	
GSDMD-Forward	GATCAAGGAGGTAAGCGGCA	195 bp
GSDMD-Reverse	CACTCCGGTTCTGGTTCTGG	
GSDME-Forward	AGTTTTCCTGGGGACTTGCT	170 bp
GSDME-Reverse	CAATGTCAGCAGAGGCAAACAA	
PJVK-Forward	TCAGCGAAGCTGACAAGTACC	300 bp
PJVK-Reverse	CCACCTCATGTTTGGTCACG	
β-actin-Forward	CCACCATGTACCCAGGCATT	253 bp
β-actin-Reverse	AGGGTGTAAAACGCAGCTCA	

## Data Availability

All data supporting the findings of this study are available within the published article. Additional materials related to this paper may be requested from the authors.

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
