# Peer review of "A Pore Forming Toxin-like Protein Derived from Chinese Red Belly Toad Bombina maxima Triggers the Pyroptosis of Hippomal Neural Cells and Impairs the Cognitive Ability of Mice"

_toxins, 2023, doi:10.3390/toxins15030191_

Round 1

Reviewer 1 Report

 The manuscript is devoted to pore-forming toxin and its properties.

Manuscript has several serious points that should be corrected and a number additional experiments should be performed to improve the manuscript.

1.     Fig.2. The data of repeated experiments (n=3 is minimal) should be performed as a number of dead cells.

2.     Pyroptosis should be supported by increased secretion of IL-18 and Il-1b.

3.     The flow cytometry experiment does not prove the interaction of βγ-CAT with cell membrane, it shows the binding of the βγ-CAT to the cell surface. To say that βγ-CAT interacts with cell membrane you should use some liposome-based assay with proper negative control to βγ-CAT. The quantification of βγ-CAT levels on the cell surface after different incubation times with the toxin can also give information about βγ-CAT endocytosis kinetics.

4.     The FITC and Cy3 emission spectras are overlapping, so the authors should provide the exact range for signal collections in two channels and proper negative control to prove the absence of the spillover.

5.     The quantification of co-localization coefficients (Pearson's or Manders') are required to prove the βγ-CAT endocytosis.

6.     Cross through BBB should be demonstrated in vivo. For example, by using fluorescently labeled βγ-CAT.

7.     It is not clear why the authors decided that increased time in the center of arena in the open filed test is associated with impaired cognitive function. Usually, increased time in the center of arena says about decreased anxiety.

8.     English is poor.

9.     Line 204: the mass of βγ-CAT is 72 kDa, while in the blot (Fig. 4B) the mass is 40 kDa.

Author Response

Thanks for your constructive suggestions. According to your suggestions, we have completed many experiments and supplemented relevant data to better support the conclusion of the paper. For example, the western blotting assay and enzyme-linked immunosorbent assay of IL-1β and IL-18 were added for better supporting the βγ-CAT triggered pyroptosis. The in vivo assay was performed and immunofluorescence experiments on brain tissue to further confirm the penetration ability of βγ-CAT on BBB. In addition, according to your suggestion, the fluorescence quantification of the flow cytometry and confocol were performed and the corresponding data have been supplemented in the revised manuscript. Finally, the English language of the manuscript has also been edited and polished by the professional language editing agency (MDPI, English editing ID:  english-61353). All these changes can be found in the revised manuscript with track changes. Thank you again for your suggestions.

Reviewer 2 Report

In this study, the toxic effects of βγ-CAT, the first naturally occurring ALP identified from a vertebrate, were investigated. Although the molecular mechanism of its action is still unclear, the authors were able to demonstrate that: βγ-CAT showed strong cytotoxicity to hippocampal neuronal cell HT-22 and triggered GSDME-dependent pyroptosis of HT-22 cells. βγ-CAT-induced pyroptosis of HT-22 cells relied on membrane binding, oligomerization and endocytosis of βγ-CAT. βγ-CAT can cross the BBB and affect the cognitive function of mice. The experimental procedure and the results appear clear and sustainable. The document in my opinion can be accepted in its current form.

Author Response

Thanks for your comments. We are delighted to hear of your appreciation of our work.  Next we will improve the format of this paper according to editors’ suggestion and the requirements of the journal. Thanks again.

Reviewer 3 Report

The mechanism of action of natural toxins in human cells is of considerable interest due to pharmacological potential. This MS analysed the action of a toad pore-forming toxin βγ-CAT on murine hypocampus cells in culture (mouse hippocampal neuronal cell line HT-22) and found evidence that hippocampal cognition was down regulated by this toxin when injected systemically into mice.

The Introduction states that βγ-CAT has diverse toxic effects on mammalian cells. The interaction between βγ-CAT and acidic glycosphingolipids may well limit the target range to neuronal cells but a comment on the cell type specificity of βγ-CAT action would be helpful.

Section 2.2. Cleavage of gasdermin D was shown to be associated with βγ-CAT treatment (Fig. 3D) and thus pyroptosis was inferred but a clearer argument or possibly more data is needed to justify that gasdermin D cleavage is necessary for pyroptosis. The membrane binding and endocytosis studies were consistent with an active role for Gasdermin D.

The mouse studies showed that βγ-CAT lowered cognition but connection to the in vitro work with HT-22 remains somewhat weak because the gasdermin profiles of other brain tissues were not shown.

Peptide toxins can be complex mixtures. What is the % purity of the βγ-CAT used in this study?

What medium was used to administer βγ-CAT to cultured cells?

Please state the number of replicates used in Figs 1, 3B, 5A and 5B.

Author Response

Thanks for your constructive suggestions. We have revised the manuscript point by point according to your suggestions. The detailed response attached below. Thanks again.

Round 2

Reviewer 1 Report

The manuscript can be accepted in the present form